# The Pathophysiology, Identification and Management of Fracture Risk, Sublesional Osteoporosis and Fracture among Adults with Spinal Cord Injury

**DOI:** 10.3390/jpm13060966

**Published:** 2023-06-08

**Authors:** Beverley Catharine Craven, Christopher M. Cirnigliaro, Laura D. Carbone, Philemon Tsang, Leslie R. Morse

**Affiliations:** 1KITE Research Institute, 520 Sutherland Dr, Toronto, ON M4G 3V9, Canada; 2Faculty of Medicine, University of Toronto, Medical Sciences Building, 1 King’s College Cir, Toronto, ON M5S 1A8, Canada; 3Department of Veterans Affairs Rehabilitation, Research, and Development Service, Spinal Cord Damage Research Center, Bronx, NY 10468, USA; 4Department of Medicine: Rheumatology, Medical College of Georgia, Augusta University, 1120 15th St, Augusta, GA 30912, USA; 5Department of Rehabilitation Medicine, University of Minnesota, 500 Harvard St SE, Minneapolis, MN 55455, USA

**Keywords:** fractures, osteoporosis, spinal cord injuries, rehabilitation, drug therapy, dietary supplements

## Abstract

Background: The prevention of lower extremity fractures and fracture-related morbidity and mortality is a critical component of health services for adults living with chronic spinal cord injury (SCI). Methods: Established best practices and guideline recommendations are articulated in recent international consensus documents from the International Society of Clinical Densitometry, the Paralyzed Veterans of America Consortium for Spinal Cord Medicine and the Orthopedic Trauma Association. Results: This review is a synthesis of the aforementioned consensus documents, which highlight the pathophysiology of lower extremity bone mineral density (BMD) decline after acute SCI. The role and actions treating clinicians should take to screen, diagnose and initiate the appropriate treatment of established low bone mass/osteoporosis of the hip, distal femur or proximal tibia regions associated with moderate or high fracture risk or diagnose and manage a lower extremity fracture among adults with chronic SCI are articulated. Guidance regarding the prescription of dietary calcium, vitamin D supplements, rehabilitation interventions (passive standing, functional electrical stimulation (FES) or neuromuscular electrical stimulation (NMES)) to modify bone mass and/or anti-resorptive drug therapy (Alendronate, Denosumab, or Zoledronic Acid) is provided. In the event of lower extremity fracture, the need for timely orthopedic consultation for fracture diagnosis and interprofessional care following definitive fracture management to prevent health complications (venous thromboembolism, pressure injury, and autonomic dysreflexia) and rehabilitation interventions to return the individual to his/her pre-fracture functional abilities is emphasized. Conclusions: Interprofessional care teams should use recent consensus publications to drive sustained practice change to mitigate fracture incidence and fracture-related morbidity and mortality among adults with chronic SCI.

## 1. Introduction

Spinal cord injury (SCI) results in a myriad of motor, sensory and autonomic impairments [1]. The individual’s neurological level of injury and American Spinal Cord Injury Association (ASIA) Impairment Scale (AIS) [2] category predict neurological and functional recovery [3], with a lower frequency of AIS category conversion or motor recovery in those with AIS A injuries versus those with AIS category C or D. The presence of comorbid infections, specifically pneumonia [4], wound infection [4] or urinary tract infection with sepsis [5], contributes to variation in functional outcomes. Further, the AIS category predicts the degree of muscle atrophy from disruption in nerve activation or denervation and muscle atrophy related to disuse and changes in muscle fiber types [6] to a predominance of type II fast-twitch fibers. Inflammatory stress and oxidative stress are the main mechanisms of disuse muscle atrophy after SCI [7]. Wheelchair use, having motor complete paraplegia, with absent or limited lower extremity voluntary muscle function and no spasticity, is associated with lower muscle mass and higher lower extremity muscle density [8]. Muscle atrophy contributes to a reduction in metabolic rate and an increased risk of developing type II diabetes mellitus. SCI is also associated with a reduction in growth hormone, and Insulin-like growth factor 1 (IGF-1), deficiencies likely to exacerbate the further loss of muscles and bone [9]. The fates of muscle and bone are linked over the lifetime of an individual with SCI.

The enclosed review will assist clinicians in preventing lower extremity fractures or treating lower extremity fractures once they occur among adults with chronic SCI or disease (more than two years post injury/disease onset) and established low bone mass or osteoporosis and a moderate or high fracture risk. A succinct review of the etiology of the acute and subacute changes in the bone mass and bone architecture of the hip, distal femur (DF) and proximal tibia (PT) regions early after acute traumatic SCI is provided for context, acknowledging that changes in bone mass and bone architecture after non-traumatic SCI are poorly described in the literature. Although guidelines regarding the prevention of sublesional osteoporosis in the subacute setting [10] exist, this topic is beyond the scope of this review. The latter sections of this review (Section 3, Section 4 and Section 5) address how to assess fracture risk, diagnose osteoporosis and select safe and effective therapy to augment bone mass and practically or theoretically reduce fracture risk, recognizing that no study to date has had fracture risk reduction or fracture incidence as a primary outcome. Clinical considerations when selecting drug or rehab therapy for the treatment of osteoporosis or optimal fracture management and criteria for stopping ineffective therapy are discussed. The authors acknowledge that the majority of the data that comprise the scientific underpinning of the consensus documents summarized herein [10,11,12] are based on data stemming from patients with traumatic SCI and may not be universally applicable to patients with SCI or disease. Readers are encouraged to read this review and the related consensus documents [10,11,12] in entirety to have a full appreciation of the concepts and nuances of practice. Health services to prevent fracture among adults with chronic SCI are optimally provided by an interprofessional team with effective communication skills and a strong emphasis on patient and family caregiver education and shared decision making.

## 2. Overview of Bone Tissue and Remodeling

### 2.1. Pathophysiology of Osteoporosis in SCI

The early regional changes in bone mass and bone architecture after traumatic SCI result in a lifetime-increased propensity for lower extremity fracture. The etiology of these changes are multifactorial and include changes in mechanical loading and strain on bone, bone turnover and blood flow. Possible influential non-mechanical factors may include poor nutritional status, hypercortisolism (either therapeutic or stress related), hormone status or alterations in gonadal function, endocrine disorders and neural factors [13].

Bone tissue is in a state of constant remodeling, which is a highly complex process that is necessary for skeletal adaptation from external loading as well as fracture healing. The Mechanostat theory, a derivative of Wolff’s law, states that strains within bone are kept within certain limits by adding and removing bone tissue, resulting in improved bone strength according to the forces that are imposed. However, if force is applied below a certain set point, bone tissue will ultimately be lost [14]. This law is clearly dominant when studying SCI as a model of immobilization osteoporosis secondary to paralysis. At the time of SCI, the process of bone loss is initiated primarily from the unloading of the skeleton, with the magnitude of bone loss in proportion to the time and degree to which the forces of ambulation on long bones are removed.

Contrary to the occurrence of fractures in postmenopausal osteoporosis, a condition in which fractures predominantly occur at the wrist, femoral neck, hip and lumbar spine, in persons with chronic SCI, the epiphyses of the DF and PT are the most vulnerable anatomic regions to fracture [15,16]. Histomorphometric studies from bone biopsies immediately after SCI have revealed an increase in osteoblast and osteoclast activity, which shifts to an increase in osteoclastic activity and a suppression of osteoblastic activity [17,18]. This suppression of osteoblastic activity is present for the first few months after SCI and returns to pre-injury levels with continued osteoclastic activity and bone resorption noted in the chronic phase of SCI [19,20]. This uncoupling of the osteoblast/osteoclast relationship is supported by the clinical findings of hypercalcemia and hypercalciuria and dramatically elevated markers of bone resorption [21].

This depressed bone turnover leads to a rapid loss in bone mineral density (BMD) within the first two years after SCI [22,23] and the deterioration of the trabecular lattice in the chronic phase of injury [24]. Several imaging studies using dual energy X-ray absorptiometry (DXA) have demonstrated BMD values in the DF and PT regions, which are composed predominantly of trabecular bone, to lose as much as 1% per week in the initial weeks to months after paralysis, resulting in a BMD loss of 50% at these sites over the first two years [22,25], and into the chronic phase of SCI [26,27]. Contrasted against other well-recognized conditions of precipitous bone loss, the rate of bone loss after acute SCI is considerably higher than what has been documented in postmenopausal osteoporosis (3–5% annually) [28], long-term bed rest (0.1% per week) [29] and space flight (0.25% per week) [30]. In addition to the demineralization and deterioration of the trabecular lattice that results from SCI, age-related changes in the bone matrix proteins and the bone matrix structure can ultimately modify bone geometry to compensate for decreases in BMD [31]. These changes in bone geometry can lead to an increase in endosteal resorption (decreased cortical thickness), with a corresponding increase in bone diameter to maintain resistance to bending and torsional forces. The thinning of the endocortical envelope, with a corresponding increase in the periosteum to preserve total bone strength, is a finding from several peripheral quantitative computed tomography (pQCT) studies in persons with chronic SCI [32,33].

To date, there is a paucity of information regarding the molecular mechanisms underpinning the pathophysiology of osteoporosis; however, recent evidence suggests some plausible models. The remodeling of bone requires the resorption of bone by osteoclasts and the formation of new bone by osteoblasts. Thus, the initiation of the remodeling process requires the maturation and proliferation of osteoclasts [34]. Osteoclasts arise from the monocytic cell lineage and are stimulated by the release of receptor activator of nuclear factor-κꞵ ligand (RANKL), with RANKL produced by osteocytes after they become encased in bone. Cells from the osteoblast lineage also release an inhibitor of RANKL known as osteoprotegerin (OPG), which increases bone density by decreasing bone resorption. In summary, RANKL drives bone resorption and stimulates osteoclastogenesis after binding RANKL to its receptor RANK, present in the membrane of osteoclast precursors, while OPG, released by osteoblasts, acts as a “decoy” receptor down regulating osteoclast activity and function. Osteoblasts are derived from mesenchymal stem cells that have morphological characteristics of protein-synthesizing cells that are well recognized for their role in bone formation [35]. Primarily synthesized by osteocytes and essential for osteoblastogenesis is the Wingless (Wnt)/ꞵ-cantenin signaling pathway and its antagonists, sclerostin and Dickkopf (Dkk-1). The Wnt signaling pathway, when coupled with mechanical loading, stimulates the osteoblast to secrete collagenous and noncollagenous proteins such as osteocalcin, osteonectin and bone morphogenic proteins that are responsible for the mineralization of the bone matrix [34].

Several studies have demonstrated the changes in the RANKL/OPG system and Wnt signaling pathway after SCI. Work performed in a rodent SCI model suggests that there is a several-fold increase in RANKL expression within 56 days of acute SCI while the expression of OPG is unchanged, making the ratio of RANKL to OPG highly unfavorable after acute traumatic SCI. In this animal model, there was also an almost 2-fold increase in osteoclast differentiation markers; in contrast, osteoblast differentiation markers were markedly depressed [36]. The few human studies that have studied the RANKL/OPG system have reported that after adjusting for age, compared to the ambulatory group, OPG concentrations were significantly lower in those with cervical non-ambulatory SCI, supporting this unfavorable relationship in humans with SCI, as lower OPG concentrations would promote bone resorption and lower BMD values over time [37]. Supporting the findings of an adverse RANKL/OPG environment, in a prospective observational study performed by Gifre et al. [38], the effect of recent SCI (<6 months from SCI) on the serum levels of RANKL and OPG and the evolution of BMD in 23 patients with SCI and 27 healthy AB controls was determined. Compared to the AB control group, the authors found that the serum levels of RANKL were significantly higher in the SCI group at baseline with additional increases at the 6-month follow-up assessment. In addition to evidence of an unfavorable RANKL/OPG ratio causing increased bone resorption in persons with SCI, the effect of mechanical unloading on the Wnt/ꞵ-cantenin signaling pathway as a result of SCI has also been a topic of recent reports on human studies.

In a study by Morse et al. [39], sclerostin levels were found to be significantly lower in persons with chronic SCI (n = 39) when compared to an age-adjusted control group (n = 10) and was the only circulating biomarker of bone resorption and formation significantly correlated with the leg bone mineral content and BMD of the DF. Lower sclerostin levels were representative of maximal unloading and bone loss in the chronic SCI cohort and support the viability of sclerostin as a biomarker to diagnose bone loss in persons with SCI when more advanced densitometric methods are not available. This finding is supported by other studies that show sclerostin levels are highest in subjects with short-term SCI (≤5 years) and decrease significantly over the first 5 years post-injury as a result of the dramatic effect of mechanical unloading on sclerostin levels, an elevation that is sustained for as long as 5 years after injury [40]. In another study by Gifre and colleagues [41], the authors compared Wnt/ꞵ-cantenin signaling antagonists, sclerostin and Dkk-1, in a subacute SCI cohort (n = 42) to an age-matched control group (n = 10), with additional follow up 6 and 12 months after the initial assessment in the SCI group. Contrary to these findings of lower sclerostin levels in the chronic SCI cohort reported by Morse et al. [39], in this subacute SCI group, sclerostin levels were similar to those in the AB controls but the Dkk-1 levels were significantly higher. Furthermore, while sclerostin levels were relatively unchanged 6 and 12 months after the initial assessment, the concentrations of the antagonist Dkk-1 increased significantly a few months after SCI and remained elevated over the course of the study. Collectively, these findings provide some insight into the complexity of pathways influencing bone remodeling in patients with spinal cord injury and low bone mass.

### 2.2. Indexes of Bone Turnover after SCI

As a result of the invasiveness and difficulty of performing bone histomorphometry studies to measure bone formation and resorption in humans, more sensitive and specific biomarkers of bone resorption and formation have been developed, with several studies performed in persons with SCI. In an early cross-sectional report by Pietschmann et al. [42], the levels of osteocalcin (OC), a marker of bone formation, were in the low normal range in 41 SCI patients 1 month after injury that increased to a peak value 7 months after SCI. In addition to OC, another biomarker of bone formation, Procollagen 1 Intact N-terminal Propeptide (P1NP), followed the same trend as that of OC where a transient depression was followed by elevation back to the normal range [43]. Contrary to the markers of bone formation that demonstrated a temporary and mild suppression soon after SCI, C-terminal cross-linking telopeptide of type I collagen (CTX-1) and N-terminal cross-linking telopeptide of type I collagen (NTX-1), biomarkers of bone resorption, were dramatically elevated outside of the normal range and remained elevated in a majority of patients with chronic SCI [20,21].

### 2.3. Calcium Metabolism and Systemic Factors Regulating Bone Metabolism

During the acute SCI phase, where skeletal resorption is the greatest (calcium released from bone), serum ionized calcium levels are elevated with an increase in the renal clearance of calcium (hypercalciuria and a high risk of renal stones). The hypercalciuria observed during the acute phase of injury begins within days of SCI and may stay elevated for several months after injury, returning to baseline levels within one year [44]. As a result of elevated ionized and serum calcium levels, there is a suppression of parathryroid hormone (PTH) release and the reduced conversion of 25 (OH) vitamin D (25 OH D) to the more biologically active 1,25-dihydroxyvitamin D (1,25(OH)2D), ultimately reducing intestinal calcium absorption [45]. During this acute period, calcium intake is often restricted by health care professionals due to the erroneous belief that increased calcium intake will increase the risk of renal stones. In chronic SCI, as in the general population, the secretion of PTH and the increase of circulating 1,25(OH)2D are subject to control by negative feedback mechanisms related to the level of serum calcium, which is, in turn, influenced by 25(OH)D levels [34].

Additional systemic factors that are important to normal bone metabolism include androgens and estrogens, which have particular significance in persons with chronic SCI. It is appreciated that androgens have a pivotal role in regulating bone homeostasis by inhibiting the osteoblastic release of local stimulating factors for osteoclastogenesis. There have been several reports of relative or absolute androgen deficiency in small cohorts of men with SCI [46,47], with a more recent retrospective analysis by Bauman et al. [48] demonstrating the prevalence of low total testosterone levels in 243 healthy men with chronic SCI. The authors demonstrated a 0.6%/year decline in serum total testosterone compared to a 0.4%/year decline in men in the Massachusetts Male Aging Study. In addition, low serum testosterone levels were observed at earlier decades of life with a higher prevalence in men with SCI than in non-disabled control subjects [49,50]. The role of estrogens in bone homeostasis is well-recognized, as the decrease in estrogen at menopause in women is the primary cause of osteoporosis. Estrogen has been observed to have anti-apoptotic effects leading to increased cell survival and a decreased inflammatory response—mechanisms of importance related to the maintenance of bone mass [51]. Several studies have shown that estrogen maintains bone homeostasis by inhibiting osteoblast and osteocyte apoptosis and preventing excessive bone resorption [52,53]. The ability of estrogens to prevent bone cell apoptosis may be just as significant in the model of immobilization osteoporosis. Animal models have shown that hind limb unloading results in immediate osteocyte and osteoblast apoptosis [54], as well as increased osteoclastic bone resorption [55]. In women with SCI, serum estrogen levels have been shown to be significantly lower than those in able-bodied (AB) controls and may be attributed to bone loss years after injury [56]. When compared to estrogen-free postmenopausal women, women with SCI demonstrate significant deterioration of trabecular bone [57]. Several reviews have summarized emerging therapies including anti-sclerostin antibodies, the mechanical loading of the lower extremity with electrical stimulation of muscle and the mechanical stimulation of bone via whole-body vibration therapy [58].

In the setting of chronic spinal cord injury/disease (SCI/D), the aforementioned changes in bone turnover, bone mass and bone architecture will have occurred prior to a clinician’s or interprofessional team’s initial assessment. The patient’s presentation may also be complicated by the presence of pre-morbid metabolic bone disease unrelated to their spinal cord impairments. Section 2 reviews the diagnostic criteria for sublesional osteoporosis, serum screening and fracture risk assessment.

## 3. Screening and Diagnosis of Sublesional Osteoporosis

### 3.1. Screening for Sublesional Osteoporosis

As highlighted previously, acute SCI results in rapid declines in hip, DF and PT bone mass, resulting in sublesional osteoporosis and increased fracture risk [58,59]. Fractures after SCI are most common at the PT followed by the DF [60], increase in incidence with time since injury and are most prevalent in those with motor complete (AIS A) injury [61,62,63]. Importantly, fracture-related deformity or limb length discrepancies or the impact of prior fracture on future fracture risk may limit the ability of patients to use emerging rehabilitative technologies such as body weight support treadmills, robotic-exoskeletons and brain computer interfaces or to benefit from advances in cure research.

Screening for sublesional osteoporosis is clinically indicated for all individuals with traumatic or nontraumatic SCI/D [11]. Screening includes a complete history and physical examination, laboratory evaluation to identify potentially treatable causes of secondary osteoporosis that may exacerbate bone loss (such as vitamin D deficiency) and the assessment of bone density and fracture risk by DXA or pQCT according to recently published SCI-specific position statements endorsed by the International Society for Clinical Densitometry [11].

The history and physical exam should focus on injury characteristics (date, age at injury, etiology of injury, neurological level and AIS category), nutrition status (protein, calcium and vitamin D intake), mobility and ambulatory status, lifestyle factors (physical activity including weight-bearing and loading activities, electrical stimulation, spinal cord stimulation, tobacco use and alcohol use), medication use, personal fracture history, parental fracture history, injurious fall history, menopausal status and menstrual status for women, spasticity assessment, body mass index and history of other comorbid conditions (such as kidney disease or diabetes), which may worsen the rate or severity of BMD decline [10]. Clinicians are encouraged to conduct fracture risk assessments on an annual basis or following a recent fall or incident (see Table 1).

### 3.2. Laboratory Screening

Regarding laboratory screening, vitamin D deficiency is common after SCI [64]. Because vitamin D is required for calcium absorption, vitamin D deficiency may lead to secondary hyperparathyroidism with increased osteoclastic bone resorption to liberate calcium from the skeleton, thereby maintaining adequate serum calcium levels. Similarly, hyperthyroidism [65], anemia [66], renal disease [67], liver disease [68,69] and diabetes mellitus [70,71] are all associated with bone loss or increased fracture risk. For this reason, all adults with SCI should have the following laboratory workup: serum 25-hydroxyvitamin D (25-(OH)D), complete blood cell count, ionized calcium or albumin-corrected serum calcium, phosphate, intact parathyroid hormone, creatinine (and estimated glomerular filtration rate), bone-specific alkaline phosphatase and transaminases, hemoglobin A1C, thyroid-stimulating hormone and 24-hour urine collection for calcium and creatinine excretion. The assessment of the hormonal status is also indicated for pre-menopausal women and all men. This includes the measurement of prolactin, follicle-stimulating hormone (FSH), luteinizing hormone (LH) and estradiol levels for both men and premenopausal women and morning fasting bioavailable testosterone for men. Additional laboratory testing is indicated to rule out other medical conditions. Serum protein electrophoresis should be considered for individuals who present with a vertebral compression fracture of unknown etiology to assess multiple myeloma or monoclonal gammopathy, a 24-hour urinary cortisol/overnight dexamethasone suppression test may be considered if Cushing’s disease is suspected and anti-tissue transglutaminase immunoglobulin A antibody levels may be measured if celiac disease is suspected. In addition to the history, physical exam and laboratory screening, bone density should be measured to screen for sublesional osteoporosis according to SCI-specific testing protocols [11].

### 3.3. Diagnostic Imaging for Sublesional Osteoporosis

Bone density may be assessed via established and reported pQCT protocols [11]. However, DXA is the current clinical “gold standard” based on cost, availability and evidence-based standards for osteoporosis diagnosis and fracture risk prediction. Because the prevalence of osteoporosis is much greater in individuals with SCI compared to the general population, irrespective of age, race, sex, or ambulatory status, all adults with SCI should have a baseline DXA of the total hip, DF and PT as soon as feasible after injury. The lumbar spine is not a recommended site for DXA assessment based on the technical difficulties associated with accurately determining bone density due to degenerative changes within the posterior elements, heterotopic ossification and/or hardware in the spine region of interest [72,73].

DXA testing should be repeated based on clinical indication such as a change in the individual’s functional abilities, interval fracture or at 1–2-year intervals to monitor response to therapy. SCI-specific protocols have been developed and previously reported for bone density determination by DXA, including acquisition protocols and normative BMD data for the DF and PT [74]. When possible, serial bone density measurements should be obtained on the same DXA machine with the same operational software and with a reported least significant change calculation for the DXA operator.

Once bone density is determined by DXA, osteoporosis is diagnosed in SCI according to the World Health Organization’s definitions using age-appropriate reference BMD to calculate T-scores or Z-scores (reported as standard deviations from the population mean). For post-menopausal women and men aged 50 or older, a T-score < −2.5 at any skeletal site tested (hip, DF or PT) is diagnosed as osteoporosis. For premenopausal women and men under the age of 50, a Z-score < −2 is diagnostic of low bone mass or secondary osteoporosis due to SCI. Additionally, an osteoporotic fracture that occurs after SCI, irrespective of bone density, is diagnostic of osteoporosis. Diagnostic criteria for sublesional osteoporosis are summarized in Table 1.

**Table 1 jpm-13-00966-t001:** Definition of sublesional osteoporosis (SLOP).

Age Range	Definition
Men ≥ 50 years or postmenopausal women	Hip or knee region T score ≤ −2.5
Men < 49 years or premenopausal women	Hip or knee region Z score < −2.0 with ≥3 risk factors for fracture
Men or women age 16–90	Prior fragility fracture and no identifiable etiology of osteoporosis other than SCI

Reprinted with permission from Topics in Spinal Cord Injury Rehabilitation; American Spinal Injury Association [75].

## 4. Treatment of Osteoporosis in SCI

### 4.1. Vitamin D Supplements

Vitamin D is a fat-soluable vitamin that helps the patient’s body absorb and retain calcium and phosphorus. An individual’s serum levels of vitamin D are dependent on the gut’s ability to absorb dietary fat [76]. Comorbid liver disease, celiac disease, Crohn’s disease and ulcerative colitis can impede vitamin D absorption [77]. Vitamin D can be found in cod liver oil, salmon, tuna, mackerel, sardines, eggs, mushrooms, milk and dairy products or orange juice fortified with vitamin D [78]. Most patients with chronic SCI cannot obtain an adequate vitamin D serum level from food sources. Vitamin D3 supplements (cholecalciferol) are usually required to ensure an adequate vitamin D intake [79].

Vitamin D insufficiency is common among those with SCI [80,81,82,83]; some estimate the prevalence of vitamin D insufficiency to be as high as 93%. A serum level of 50 nmol/L defines vitamin D deficiency, whereas a level of 75 nmol/L defines vitamin D insufficiency [84]. Low vitamin D (25-(OH)D) levels are associated with low serum testosterone levels [85], poor immune and respiratory system function, impaired balance and low leisure time physical activity [86]. There is a small amount of data regarding the optimal levels of vitamin D for bone and muscle health in individuals with chronic SCI. A validated assay should be used [87,88,89,90,91,92,93,94,95] to assess the vitamin D serum levels of adults with chronic SCI.

Vitamin D maintenance therapy with 1000–2000 IU or 25–50 mcg of vitamin D3 (cholecalciferol) per day is recommended [79,96], assuming that vitamin D serum levels are adequate. A reasonable target for a 25-(OH) D level is 100 nmol/day (40 ng/mL) [79,97]. Bauman et al. [88,98] reported that 2000 IU of vitamin D supplementation raised 25-OH vitamin D serum levels among individuals with chronic SCI.

### 4.2. Dietary Calcium and Nutritional Considerations

An adequate calcium intake is important to ensure patients with chronic SCI maintain their lower extremity BMD and reduce their fracture risk. A majority of patients with SCI (≈72% of one study) do not meet the Dietary Reference Intake for Calcium [99]. Patients with SCI and neurogenic bowel dysfunction often need to consume dairy and non-dairy sources of calcium to achieve an adequate intake. Clinicians are encouraged to have their patients complete a food frequency questionnaire such as the Calcium Assessment Tool [100] and to review their patient’s cumulative calcium intake through supplements and diet prior to making recommendations or initiating dietary counselling. An individual’s calcium intake should preferably be through diet, as opposed to supplements, and efforts should be made to ensure the patient’s intake meets the Dietary Reference Intake [101]. An individual’s recommended dietary calcium intake depends on his/her age, sex and history of renal and bladder stones (Table 2).

Individuals with SCI, neurogenic bladder and calcium oxalate stones should have a lower calcium intake of 750 mg/day from food and/or supplements. Patients with chronic SCI taking vitamin D may experience hypercalciuria and are at an increased risk of developing renal stones. Patients with chronic SCI frequently have recurrent stones, staghorn calculi and bilateral stone disease [102]. The cumulative proportion of patients with chronic SCI who had renal calculi was 38% in a longitudinal study of 45 years [103]. An annual screening ultrasound of the bladder and kidneys can detect asymptomatic stone disease, particularly in patients with frequent urinary tract infections.

Calcium apatite and oxalate stones are the most prevalent. SCI clinicians should work to differentiate calcium oxalate and hydroxyapatite stones [104] among patients with SCI. Some SCI patients may need to adhere to a low oxalate diet to reduce their stone disease risk. Foods to avoid that are high in oxalates include spinach, cranberries, rhubarb, beets, beans, berries, chocolate or coffee.

### 4.3. Fracture Risk and Rehabilitation Therapy

Prior to initiating rehabilitation therapy, clinicians should review with their patients any concurrent medical therapies with potential adverse events on bone density/quality and non-BMD risk factors for fracture (see Table 3, e.g., alcohol intake). Prior to patients with chronic injury engaging in rehabilitation therapy that requires loading, several factors should be considered by the clinician, including the patient’s hip, knee and ankle region range of motion, lower extremity alignment and BMD.

As sublesional osteoporosis progresses, there is a reduction in trabecular volume and number as well as cortical thinning. Due to these changes, the femur and tibia become more susceptible to injury after a torsional stress or non-anatomic loading. There are currently no established thresholds of BMD that absolutely contraindicate patients with SCI from participating in weight-bearing activities [11]. However, it is critical that clinicians review with their patients the possible risk of fracture when participating in rehabilitation therapy and document the consent discussion regarding the risks and benefits of the planned rehabilitation intervention. In select cases, particularly among patients with chronic SCI, clinicians may consider radiographs of the patients’ forefoot, calcaneus and ankle in order to confirm the patient’s self-report of no pre-morbid fracture history. It is critical to report fractures and other musculoskeletal injuries that occur during rehabilitation therapy, although adverse event reporting during exercise interventions has typically been under-reported, inconsistent and suboptimal [105].

### 4.4. Rehabilitation Therapy: Standing and Walking

Bone is a dynamic tissue, which responds to increases and decreases in load. Standing and walking are common rehabilitation strategies to facilitate loading following SCI, provided these tasks are functionally appropriate for the individual based on their neurologic level of injury and AIS category, the individual’s therapeutic goals and the plan of care [106]. Passive standing, in which muscle activation is unlikely, may be performed with individuals with SCI in a standing frame, standing wheelchair, long leg braces, or other devices [10]. Passive standing for 1 h, five times per week, may reduce BMD decline, specifically in the hip and knee regions [107]. However, standing with an orthosis must be done longitudinally to measure the effects on lower extremity BMD. The high percentage of the force applied through the upper limb and shoulder when using forearm crutches (30–50% of body weight) and the energy cost of donning and doffing lower extremity orthoses or transferring in and out of a standing frame are important feasibility constraints that may impede longitudinal passive standing adherence [108].

Walk training may begin on a treadmill and can include supportive harnesses, orthoses, exoskeletons [109], or assistive devices [110] before progressing to overground walking with or without a gait aid. Despite the obvious progressions in loading available through the provisions of harnesses, orthoses/exoskeletons and supportive gait aids, walking over ground or on a treadmill is an insufficient stimulus to maintain or improve low bone mass in patients with chronic SCI. Walking [111] is a recommended therapeutic goal to maintain or increase bone strength in the general population [112]. Although the therapeutic value of walking as a rehabilitation intervention to improve functional independence and the patient’s physiological health is acknowledged [108], there is weak evidence and significant methodological limitations in treadmill, orthoses and walking training interventions for improving lower extremity BMD in the setting of chronic SCI [113,114,115].

### 4.5. Rehabilitation Therapy: Neuromuscular Electrical Stimulation (NMES) and Fucntional Electrical Stimulation (FES)

The recently published Paralyzed Veterans of America (PVA) guidelines distinguish neuromuscular electrical stimulation (NMES) and functional electrical stimulation (FES). NMES is defined as, “the application of an electrical current of sufficient intensity to elicit muscle contraction”, whereas FES refers to,”the process of pairing NMES simultaneously or intermittently with a functional task, such as cycling or rowing”. NMES and FES have the potential to mechanically load bone and influences bone mass and bone structure, and muscle contractions can produce physiological loads on bone in order to activate the muscle bone unit [36] through Wnt signalling molecules, Receptor activator of nuclear factor kappa-Β ligand (RANKL) and Osteoprotegerin (OPG). NMES or FES is recommended for treating patients with low bone mass or osteoporosis of the lower extremities in persons with a SCI [116,117]. FES cycling or FES standing for 30 mins per day or 3–5 h per week over one year may improve skeletal health [118,119,120,121]. Effective NMES or FES should (1) create a visibly strong contraction against incremental resistance when doing an isometric contraction, moving against gravity or weight bearing, (2) use a pulse duration of 200 us or higher and (3) use frequencies of 20–33 Hz and (4) amplitudes of up to 140 mA for at least 9 months. Ideally, rehabilitation interventions result in adaptive changes in bone architecture, which will produce reductions in fracture incidence. To maintain the effects of rehabilitation interventions on bone density, lower extremity muscle activation and load bearing need to continue in perpetuity, a major feasibility and cost constraint for most individuals. Further, there are many relative contraindications to the delivery of rehabilitation interventions among patients with chronic SCI, particularly those with a high fracture risk and prior fracture.

### 4.6. Drug Therapy: Alendronate, Zoledronic Acid and Densoumab

Along with optimizing vitamin D and calcium intake and stimulating bone formation with rehabilitation interventions, pharmacological interventions should be considered best practice for the management of patients with chronic SCI, established osteoporosis and moderate or high fracture risk. There are well-documented precipitous declines in trabecular BMD [32,122,123,124] during the first 2 years after SCI, after which BMD declines but to a far lesser degree over the remainder of the individual’s lifespan. The PVA clinical practice guidelines recommend that individuals with SCI and diagnosed low bone mass (Z-score of < −2.0) or osteoporosis (T-score of ≤ −2.5) be offered oral Alendronate, intravenous (IV) Zoledronic acid (ZA), or subcutaneous Denosumab to treat total hip, DF, or PT BMD [10]. After a discussion of the merits and potential adverse effects, the patient may want to pursue drug therapy to prevent further declines in hip and knee region BMD.

Although there are many oral bisphosphonates on the market, only Alendronate has been studied in the chronic SCI population [10]. Alendronate (10 mg daily or 70 mg weekly) may maintain or improve BMD and increase the cortical bone mineral content in the tibial epiphysis, tibial diaphysis, total hip [125], femoral neck, DF epiphysis, distal femoral metaphysis, PT epiphysis and spine [126]. ZA is a more potent bisphosphonate than Alendronate. ZA, given as a one-time infusion of 5 mg/100 mg solution infused over 15 min, has been reported to maintain BMD of the total hip, femoral neck and distal radius and produce increments in the cortical bone volume of the DF metaphysis and PT metaphysis when used with FES cycling [127]. Further, some studies suggest therapy with Denosumab for one year within 15 months of injury onset to increase total hip and femoral neck BMD [128]. It is important to note that the mechanisms of action and effects of bisphosphonates and Denosumab differ (Baron et al. 2011; Tu et al. 2018) [129,130], but a discussion of these pathways is not within the scope of this review.

#### 4.6.1. Safety Considerations, Absolute and Relative Contraindications to Therapy

Prior to initiating drug therapy for sublesional osteoporosis, clinicians are encouraged to use shared decision making that accounts for the patient’s allergies, comorbidities, values and risk/adverse event tolerance. A detailed discussion of the relative and absolute contraindications of specific therapies must be considered as well as the patient’s sex and age before prescribing therapy, as there are unique contraindications for females of childbearing potential and unique contraindications based on age, allergies, prior cancer, dental history and the presence or absence of lower extremity edema or pressure injury. Alendronate, ZA and Denosumab are absolutely contraindicated in pregnancy [10]. Sexually active females of childbearing age choosing to take these medications will require birth control.

An important ZA relative contraindication is the patient’s renal function. Patients whose creatinine clearance is less than <30–35 mL/min should not be offered this therapy [10]. Likewise, a latex allergy, significant lower extremity edema or pressure sores [131] and hypocalcemia should be ruled out prior to prescribing Denosumab treatment [10], as Denosumab can increase the risk of lower extremity cellulitis.

Serious adverse events such as osteonecrosis of the jaw (ONJ) and atypical femoral fracture (AFF) are rare and have been reported with long-term oral or IV bisphosphonate therapy in the non-SCI population. Patients who report a prior history of cancer and radiotherapy are most like to develop ONJ [132,133]. The reported incidence of ONJ in patients taking oral or IV bisphosphonates long-term varies from 1 per 10,000 to 1 per 100,000 patient-treatment years [134]. Prior to oral bisphosphonate prescription or IV ZA administration, clinicians are encouraged to examine the patient’s oral cavity for exposed gums, broken or abscessed teeth, or gum disease.

AFF is very rare and is often characterized by a transverse stress fracture of the femoral shaft in the sub trochanteric region, usually with a prodrome of groin pain originating laterally and extending through both cortices, with localized periosteal or endosteal cortical thickening [135], without a history of trauma. AFF incidence ranges from 1.8 per 100,000 per year after 2 years of bisphosphonate exposure to 113 per 100,000 per year after 8 to 9.9 years of drug exposure [136], suggesting AFF risk increases with the duration of bisphosphonate therapy. For primary postmenopausal osteoporosis, the benefit of bisphosphonate therapy from reduced fracture risk exceeds the risk of developing either ONJ or AFF [137]. AFF will typically result in the cessation of bisphosphonate therapy [138]. Fortunately, no definitive cases of ONJ or AFF have been reported following drug administration among patients with chronic SCI [139,140,141,142,143,144,145,146].

#### 4.6.2. Monitoring and Cessation of Ineffective Therapy

The optimal duration of osteoporosis therapy for patients with chronic SCI is unknown. Prescribing drug therapy requires the treating physician to routinely assess treatment adherence and treatment effectiveness and screen for side effects. Physicians should have a low threshold for stopping ineffective therapy or therapy associated with persistent side effects. Bisphosphonate adherence is poor in general; patients with chronic SCI in the community report being on a mean of 11 (SD  =  6) classes of drugs [147]. The most common drugs prescribed are laxatives, opioids and cardiovascular-related drugs. Thus, for patients with chronic SCI and polypharmacy assessment of osteoporosis, drug therapy adherence is important before determining whether therapy has been ineffective. The most commonly report reasons for poor adherence include gastrointestinal side effects [148].

It is critical that clinicians reassess and consider stopping, continuing or changing therapy on a routine basis, particularly if significant declines in hip, DF or PT BMD occur after two years of therapy adherence or if a lower extremity fracture occurs in an individual with chronic SCI who has been adherent to therapy for at least one year.

Clinicians can use the least significant change (LSC) to detect true biological change over time, defined as a gain or loss in BMD that exceeds the LSC of the densitometer and the local DXA technologist. LSC is specific to the diagnostic tool used and its accompanying precision. In some circumstances, drug holidays may also appropriate for individuals with moderate fracture risk following 5 consecutive years of oral Alendronate or 3 years of intravenous ZA. In the context of low bone mass or osteoporosis and moderate-to-high fracture risk, there is much uncertainty regarding the optimal duration of treatment and the circumstances or BMD thresholds for changing drug therapy. In the current scenario, clinicians are encouraged to follow the consensus opinions for therapy duration for post-menopausal and steroid-induced osteoporosis in the absence of evidence, recognizing an urgent and compelling need for better evidence to recommend actions regarding treatment duration [149,150]. Alternative treatment options may involve hormone therapy, although in the current review we are unable to discuss the role of estrogen and hormone therapy [151]. Likewise, there is a need to further explore the effects of hormone therapy on the maintenance of bone mass in SCI/D populations [152].

## 5. Fracture Management

This section of the review underscores the importance of an initial orthopedic consultation to confirm the presence of a fracture and discusses the merits of conservative versus operative fracture management. Specific medical and mobility considerations to reduce fracture-related morbidity and the current rates of fracture-related mortality are discussed.

### 5.1. Fracture Incidence

Long bone fractures are common, with an annual incidence of 2–3 lower extremity fractures/100 person years [15,20,61,153]. The tibia/fibula, femur and hip are among the most common sites fractured [61,154]. The majority of these fractures result from wheelchair or transfer-related falls [60,155,156]. Other lower extremity sites, including fractures of the talus [157] and calcaneus [158], have been reported to occur while using exoskeletons [159]. Upper extremity fractures are less common, with one study reporting that approximately 83% of all long bone fractures are in the lower and only 17% in the upper extremity, with the most common upper fracture site being the humerus [160]. The risk of mortality following a lower extremity fracture is greater in men older than 50 years of age (hazard ratio [HR] 3.42, 95% confidence interval [CI] 2.75–4.25), in men with motor complete injury (HR 3.13, 95% CI 2.19–4.45) and in men with a high Charlson Comorbidity Index [161]. Women with SCI over age 50 are at a higher risk of fracture than are younger women or men of any age [HR 1.56, (95% CI 1.18–2.06)]. Both the fracture itself and chronic comorbid conditions contribute to mortality in older men with SCI.

### 5.2. Operative or Non-Operative Management of Fractures

The goal for the treatment of osteoporotic fractures, whether done operatively or non-operatively, is to return the individual to their pre-fracture level of mobility and independence [162,163]. Treatment choices should be guided by an ascertainment of the risks and benefits of operative versus non-operative approaches, the person’s premorbid level of function and consideration for the impact of fracture management strategies on the ability to use both current and future mobility technologies [12]. Regarding the latter, the effects of the fracture treatment modality on the range of motion are important to consider as current robotic exoskeleton devices require a specific range of motion for hip and knee extension and neutral ankle-joint positions [159,164,165,166,167,168,169,170,171,172]. Education on the risks and benefits of operative compared with non-operative approaches should be provided, and shared decision-making should form the basis for the fracture treatment [12]. However, the extent to which this is happening in clinical practice is not clear. In one study including Veterans with a lower extremity fracture, some felt that they would have preferred operative treatments but were not offered these [173].

Non-operative treatment of fractures includes circular [15] or bivalve [174] casts, padded [175] or pillow [174] splints, leg lengthening devices [176] plaster casts or shells [177,178], traction [15,174,175,178,179,180,181,182], external fixation [176] and observation [183]. Such non-operative strategies have traditionally been considered the treatment of choice for fractures, as historically surgical treatment of fractures was associated with high complication rates [180,184]. A CPG for the management of acute fractures concluded that if non-operative therapies are chosen, these should be well padded with attention to neutral rotation alignment and should allow for frequent skin inspection [12]. No specific recommendations for particular immobilization devices were given [12].

However, non-operative therapies for fracture treatment have risks; these largely stem from a longer time for fracture healing with potential complications from immobilization and subsequent impact on the activities of daily living [185]. In patients with lower extremity fractures in which the treatment was fracture immobilization with a splint, more than 80% were admitted for prolonged inpatient stays, principally because of inadequate support at home or inaccessibility of the home environment for the patient [186]. These concerns have led to an interest in operative strategies for fracture management. Such strategies include open reduction with internal fixation (ORIF) [15,183], intramedullary rodding and/or nailing [155,174,175,176,177,178,187,188,189], plate and screw fixation [182,188], resections [182], compression screw devices [187] and total or hemi arthroplasties [175,183,190]. Amputations are most often done after the failure of non-operative treatments [183,191].

There are a few definite indications for the primary operative management of extremity fractures. Open fractures and fractures that an orthopedic surgeon determines would not reliably heal in a position that would restore the patient to their pre-fracture function status are two such situations [12]. Other indications for surgical intervention stem from the failure of primary non-operative therapy, including the development of a nonunion or malunion associated with residual deformity that impairs functional ability. Conversion to an open fracture after primary non-operative management is another indication for surgery. Surgical treatment should also be considered for a National Pressure Ulcer Advisory Panel Stage-3 or 4 pressure injury that has failed to heal with conservative therapy [12].

ORIF has historically been the most common operative strategy for long bone fracture repair [175,181,183,189,192]. The Orthopaedic Trauma Association provides operative strategies by fracture site. These include internal fixation for nondisplaced hip and femur fractures, arthroplasty for displaced femoral neck fractures and internal or external fixation for tibia/fibula and ankle fractures. Percutaneous fixation is considered for foot fractures. In view of the increasing use of exoskeleton devices, the authors suggested that patients should avoid early return to weight-bearing activities after the healing of ankle/foot fractures because of the high risk of re-fracture with upright activities such as the exoskeleton [12].

Few reports have compared outcomes following non-operative versus operative fracture treatments, and there are no randomized clinical trials addressing this. Several studies have suggested improved outcomes following the operative management of fractures [155,177,188,189]. Some [155,177], but not all [182,193], have noted fewer fracture nonunion with operative compared with non-operative therapies. No differences in malunion rates by type of fracture treatment were noted in one report [194]. A better range of motion was achieved in a cohort managed surgically compared to those managed non-operatively [177]. Amputations resulting from fracture-related complications are more than twice as common when nonsurgical compared to surgical treatments are used as the initial management for lower extremity fractures [191]; diabetes mellitus [183,191,195] and peripheral vascular disease [191,195] are additional risk factors [191]. Mortality is increased following lower extremity fractures [161]; however, no differences in mortality based on operative or non-operative treatment have been reported [183,185].

### 5.3. Rehabilitation Post Fracture

Physical and occupational therapists and, for wheelchair users, seating specialists, should be involved in fracture care to assist with range of motion exercises, provide recommendations for braces and devices to prevent pressure injuries and assess needs for patient equipment, skills training and/or temporary increases in caregiver support [196].

Upwards of half of patients with chronic SCI and a lower extremity fracture [60,61] may develop fracture-related complications. These include infections [15,182,190], compartment syndrome from a poor-fitting device for immobilization [197], heterotopic ossification [198], pressure injuries [60,61,185,199], fracture nonunion [60,155,193], venous thromboembolism [199], pain [12], autonomic dysreflexia and spasticity [60]. The extent to which depression and anxiety occur after a long bone fracture in patients with a SCI is not clear; affective disorders commonly complicate open lower-limb fractures in the AB population [200]. For individuals with a hip, DF or PT fracture, thromboprophylaxis should start as soon as is feasible, with therapy continuing for 2–4 weeks or until the person returns to their premorbid mobility status for non-wheelchair users. Patients awaiting definitive fracture management require monitoring for signs and symptoms of autonomic dysreflexia. As shown in Figure 1, the evaluation of a patient’s BMD and fracture risk and initiation of appropriate therapy should begin as soon as feasible following definitive fracture management. Wheelchair users with a visible limb length discrepancy or change in their seated posture require seating intervention and a graded return to their premorbid wheelchair skills, while patients who walk or use gait aids who have a limb length discrepancy or residual deformity often require orthoses, functional upgrading, gait and balance training to return to their premorbid functional abilities.

Many gaps still remain in our understanding of what constitutes optimal fracture repair in persons with a SCI. Future studies should also address which patients benefit most from operative compared with non-operative fracture repair with the goal of any repair being to return to the patient to their pre-fracture functional level as quickly as possible.

## Figures and Tables

**Figure 1 jpm-13-00966-f001:**
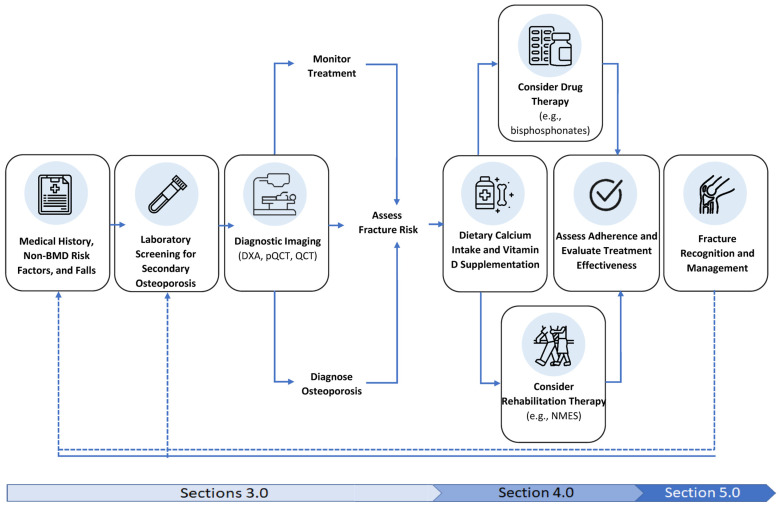
Flow diagram of the process of identifying, treating, and managing sublesional osteoporosis and fractures in SCI (Figure adapted with permission from Paralyzed Veterans of America; Consortium for Spinal Cord Medicine Clinical Practice Guidelines [10]).

**Table 2 jpm-13-00966-t002:** The following are recommendations for calcium intake as a combination of food and supplements (preference for dietary intake over supplements).

Group and Age	Calcium Recommendation
Men & premenopausal women age 19–50 years	1000 mg/day
Men 50–70 years	1000 mg/day
Women 50–70 years	1000–1200 mg/day
Men and women 71+ years	1000–1200 mg/day

Reprinted with permission from Paralyzed Veterans of America; Consortium for Spinal Cord Medicine Clinical Practice Guidelines [10].

**Table 3 jpm-13-00966-t003:** Fracture Risk Factor Checklist Prior to Rehabilitation Intervention (Adapted with permission [10]).

Established Fracture Risk Factors	
□	Alcohol intake > 5 servings per day
□	Paraplegia
□	Duration of SCI ≥ 10 years
□	Motor complete injury (AIS A–B)
□	Family history of fracture
□	Hip fracture in the last year
□	Routine use of benzodiazepines, anticonvulsants (i.e., carbamazepine, phenytoin), heparin, opioid analgesia (≥28 mg morphine for a 3-month period)

Reprinted with permission from Paralyzed Veterans of America; Consortium for Spinal Cord Medicine Clinical Practice Guidelines [10].

## Data Availability

The data presented in this review are openly available in References [8,9,10].

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
