# Peer review of "The Pathophysiology, Identification and Management of Fracture Risk, Sublesional Osteoporosis and Fracture among Adults with Spinal Cord Injury"

_jpm, 2023, doi:10.3390/jpm13060966_

Round 1

Reviewer 1 Report

The manuscript "The pathophysiology, identification, and management of fracture risk, sub lesional osteoporosis, and fracture among adults with spinal cord injury." has been prepared and is well presented. The review revolves around fracture risk and sub lesional osteoporosis in spinal cord-injured patients. An alarming rise in spinal cord injury prevalence necessitates such a review around the globe.

 There are a few major improvements I would like the authors to make to their manuscripts

1. The acronyms are listed at the end of the manuscript. Throughout the manuscript, some words are not abbreviated, and even when they are, they are not used wisely. Even though it has been abbreviated, authors are highly encouraged to abbreviate or expand the abbreviation when it appears first.

2. Section 2.1 – Pathophysiology of Osteoporosis should be detailed with its molecular basis to improve the manuscript.

3. Section 2.3 “Actions of estrogen on osteoporosis” should be detailed to improve the manuscript.

4.  Page 4 – Line No. 181 is not necessary since it has been captioned in Figure 1.

5.  Page 4 Line 182, Screening and Diagnostics is listed under section 3, but written as section 2. Kindly verify.

 6. Section 3.1 Line no. 190 to 195 seems repeated. Consider revising the statement

7. Section 3 Screening and Diagnosis of sub-lesional osteoporosis can be illustrated in a simple way to understand.

8.     Page 8 – Line 332 typo “a to torsional stress”  

 9. There is a common heading for sections 4.4 and 4.5. This section deals with NEMS

10. Section 4.6 Mechanism of action of a drug and its impact on osteoporosis can be detailed.

11. Page 4 The acronym list can be updated.

Phrases are not lucid to understand.

Reviewer 2 Report

Title: The pathophysiology, identification and management of fracture risk, sublesional osteoporosis and fracture among adults with spinal cord injury.

This review aimed to synthesize the best practices and guideline recommendations that are articulated in recent international consensus documents (International Society of Clinical Densitometry, Paralyzed Veterans of America Consortium for Spinal Cord Medicine and the Orthopedic Trauma Association) which highlight the density decline and pathophysiology of lower limbs bones minerals after acute spinal cord injury.

Main comments

In general, the manuscript is well-written with several technicalities of this field. Be careful with commas in some sentences. Some specific comments are presented below.

Specific comments

0. Abstract

- Line 14: Write “recommendations” (plural) instead of “recommendation”.

- Lines 17-18: Write “…consensus documents which highlight …” instead of “…consensus documents; which highlights…”.

- Line 24: Write “…stimulation (NMES)) to modify…” (ending bracket) instead of “…stimulation (NMES) to modify…”.

1. Introduction

- Line 37: Space between “injury” and the acronym “(SCI)”.

- Lines 49-50: Write “…contributes to a reduction in…” instead of “…contributes to reductions in…”. Same correction in Line 51.

- Lines 52-53: Write “…muscles and bones…” (plural) instead of “…muscle and bone…”.

- Line 56: Write “more than” instead of “>”.

2. Overview of Bone Tissue and Remodeling

- Lines 80-82: Write “The etiology of these changes in lower extremities are multifactorial…” instead of "The etiology of these regional changes in lower extremity bone mass and bone architecture after SCI are multifactorial…”.

- Lines 98-99: Write “…the most vulnerable anatomic regions …” instead of "…anatomic regions most vulnerable…”.

- Lines 148-150: Be careful with commas in “During this acute…renal stones”.

- Line 155: Write “…and have particular significance …” instead of "…and are of particular significance…”.

- Line 157: Delete the space before “There” at the end of the line.

- Line 164: Write “…non-disabled …” instead of "…nondisabled…”.

- Figure 1: Insert a better-quality image and without underlining mistakes to review (“pQCT”).

3. Screening and Diagnosis of Sublesional Osteoporosis

-Line 264: Write “…is diagnosed as osteoporosis” instead of “…is diagnostic of osteoporosis”.

4. Treatment of Osteoporosis in SCI

- Line 279: Space between “SCI” and the references “[70-73]”.

- Line 300: Space between “Tool” and the reference “[90]”.

- Line 307: Delete “:” at the end of the sentence.

- Line 317: Write “…to differentiate…” instead of “…to differentiation…”.

- Line 320: Write “…chocolate or coffee.” instead of “…chocolate and coffee to name a few.”.

- Lines 330-332: Be careful with commas in “As sublesional…anatomic loading”.

- Line 350: Write “devices” (plural) instead of “device”.

- Line 367: Space between “SCI” and the references “[104-106]”.

- Line 369: Write “…NMES and FES. NMES is defined…” (with “.”) instead of “…NMES and FES, NMES is defined…”.

- Line 378: “Delete 30 mins” or specify “30 mins per day”.

- Lines 400-401: I do not understand the sentence. Rewrite it in another way.

- Line 440: Delete the space between “AFF” and “incidence”. Same in Line 441 between "of” and “Bisphosphonate”.

5. Fracture Management

- Lines 498-499: Write “non-operatively” instead of “nonoperatively”. Same in Lines 507, 515, 518, 521… to maintain the homogeneity of the term.

- Line 504: Write “are” (plural) instead of “is”.

- Line 535: Space between “[12].” and “Other”.

- Lines 545-546: Write “…is considered for foot fractures.” instead of “…is a consideration for foot fractures.”.

- Line 587: Write “studies” (plural) instead of “study”.

- Line 588: Write "repair” instead of "re-pair”.

List of Acronyms: Delete (FSH) in FSH definition.

References

- Finish each reference with a “.” after DOI.

- Add DOI in references 14 and 15.

- Reference 20: Journal abbreviation is “Eur. J. Clin. Invest.”.

- Reference 36: Journal abbreviation is “N. Engl. J. Med”.

- Reference 50: Journal abbreviation is “Int. J. Phys. Rehabil. Med.”.

- References 68, 90, 91: End bracket “]” at the end of the reference.

- Reference 81: ¿Vitamin DSP is an author?

- Reference 99: Journal abbreviation is “J. Orthop. Trauma Rehabilitation”.

- Reference 103: Introduce URL and delete “[“ at the end of the reference.

- References 113, 134: Journal abbreviation is "Calcif. Tissue Int."

- Review the format of references 166-169, 180.
